# Facile Microwave Assisted Synthesis of Silver Nanostars for Ultrasensitive Detection of Biological Analytes by SERS

**DOI:** 10.3390/ijms23158830

**Published:** 2022-08-08

**Authors:** Radu Nicolae Revnic, Gabriela Fabiola Știufiuc, Valentin Toma, Anca Onaciu, Alin Moldovan, Adrian Bogdan Țigu, Eva Fischer-Fodor, Romulus Tetean, Emil Burzo, Rareș Ionuț Știufiuc

**Affiliations:** 1Department of Family Medicine, “Iuliu Hatieganu” University of Medicine and Pharmacy, 2-4 Clinicilor Street, 400006 Cluj-Napoca, Romania; 2Faculty of Physics, “Babes-Bolyai” University, 1 Kogalniceanu Street, 400084 Cluj-Napoca, Romania; 3Department of BioNanoPhysics, MedFuture Research Center for Advanced Medicine, “Iuliu Hatieganu” University of Medicine and Pharmacy, 4-6 Pasteur Street, 400337 Cluj-Napoca, Romania; 4Department of Pharmaceutical Physics & Biophysics, Faculty of Pharmacy, “Iuliu Hatieganu” University of Medicine and Pharmacy, 6 Pasteur Street, 400349 Cluj-Napoca, Romania; 5Department of Translational Medicine, MedFuture Research Center for Advanced Medicine, “Iuliu Hatieganu” University of Medicine and Pharmacy, 4-6 Pasteur Street, 400337 Cluj-Napoca, Romania; 6Oncology Institute “Prof. Dr. Ion Chiricuta”, 400015 Cluj-Napoca, Romania

**Keywords:** anisotropic silver nanostars, SERS, bioanalytes, cell lysates, pharmaceutical compounds

## Abstract

We report a very simple, rapid and reproducible method for the fabrication of anisotropic silver nanostars (AgNS) that can be successfully used as highly efficient SERS substrates for different bioanalytes, even in the case of a near-infra-red (NIR) excitation laser. The nanostars have been synthesized using the chemical reduction of Ag^+^ ions by trisodium citrate. This is the first research reporting the synthesis of AgNS using only trisodium citrate as a reducing and stabilizing agent. The key elements of this original synthesis procedure are rapid hydrothermal synthesis of silver nanostars followed by a cooling down procedure by immersion in a water bath. The synthesis was performed in a sealed bottom flask homogenously heated and brought to a boil in a microwave oven. After 60 s, the colloidal solution was cooled down to room temperature by immersion in a water bath at 35 °C. The as-synthesized AgNS were washed by centrifugation and used for SERS analysis of test molecules (methylene blue) as well as biological analytes: pharmaceutical compounds with various Raman cross sections (doxorubicin, atenolol & metoprolol), cell lysates and amino acids (methionine & cysteine). UV-Vis absorption spectroscopy, (Scanning) Transmission Electron Microscopy ((S)TEM) and Atomic Force Microscopy (AFM) have been employed for investigating nanostars’ physical properties.

## 1. Introduction

Plasmonic nanoparticles are in the limelight of modern nanotechnology applications, basically owing to their unique optical properties that are strongly dependent on their shape and size. The possible use of plasmonic nanoparticles as highly effective Surface Enhanced Raman Spectroscopy (SERS) substrates when using a NIR excitation laser led to the development of a plethora of synthesis methods for different types of noble metal nanoparticles, both isotropic or anisotropic [1,2]. Over time, it turned out that spherical isotropic plasmonic nanoparticles have a limited applicability as SERS substrates given their relatively low electromagnetic field enhancement. As a result, different aggregation techniques, leading to the creation of randomly oriented hot-spots capable of locally amplifying the Raman spectra of the analytes have been reported in the literature [3].

From the point of view of analytical quantitative analysis, the aggregation technique holds a major drawback since the process cannot be controlled and the amplified Raman spectra of analytes lack reproducibility [4,5]. The random orientation of the hot-spots and the very poor reproducibility of the amplified Raman spectra represent the two most important obstacles that must be overcome for the introduction of SERS spectroscopy in clinical ultrasensitive biodetection applications.

On the other hand, it has been demonstrated that the strong Surface Plasmon Reso-nance (SPR) effect observed in the case of anisotropic nanoparticles significantly influ-ences the SERS effect [6,7]. SERS enhancement depends on the size but mostly on the shape of the plasmonic substrates. For instance, in the case of anisotropic silver nanoparticles, it has been shown that the amplification can be 20 times higher than that of silver nanospheres [8].

In this paper, we present a very simple and rapid method for the creation of a new class of SERS substrates able to generate highly reproducible SERS spectra for molecules of biomedical interest, possessing a very low Raman cross section (e.g., pharmaceutical compounds, cell lysates). The here-proposed plasmonic substrates have the capacity to operate even in the case of using a NIR excitation laser, which represents a sine qua non condition for SERS measurements performed on bio analytes. The plasmonic substrates consist of silver nanostars, synthesized using an original method developed in MedFuture’s Research Center laboratory. The only chemical precursors employed in this wet-chemical procedure are silver nitrate and trisodium citrate, with no addition of any other reagents. The microwave-assisted synthesis of the nanostars has been performed in a sealed bottom flask, which turned out to be a critical parameter in the synthesis process. The chemical reaction took place inside of a microwave oven capable of generating an almost uniform temperature in the whole volume of the liquid in a very short period. By the end of the synthesis process, the colloidal solution was immersed in a water bath at 35 °C. The SERS activity of the here-proposed AgNS has been evaluated on test molecules (Methylene Blue) as well as on pharmaceutical compounds with low Raman cross section (doxorubicin, atenolol & metoprolol), cell lysates and amino acids (cysteine & methionine), by pouring a very small volume of colloidal solution (~1.5 µL) onto a ${\mathrm{MgF}}_{2}$ glass plate heated at 50 °C, prior to the deposition of the analytes. The quality of the vibrational spectra collected using the here-proposed plasmonic substrates represents strong experimental evidence of their versatility, especially in the case of bioanalytes.

## 2. Results and Discussion

In this study we proposed an original procedure for the microwave-assisted synthesis of silver nanostars by involving trisodium citrate as a reducing and stabilizing agent of ${\mathrm{Ag}}^{+}$ ions. The role of microwave heating is to create a more homogenous thermal environment for the synthesis procedure and to speed up the process as shown in literature [9]. Once the synthesis procedure was completed, the nanostars’ morphology was analyzed by microscopic complementary techniques (TEM/STEM, AFM). Their plasmonic properties have been tested on several biological analytes by means of SERS using a NIR excitation wavelength (785 nm). The SERS experiments were carried out on interconnected AgNS that can be regarded as solid plasmonic substrates, as well as on individual nanostars that can be easily localized on the ${\mathrm{MgF}}_{2}$ port-probe using an optical microscope.

### 2.1. Characterization of AgNS Colloids by UV-Vis Spectroscopy

UV-Vis absorption spectroscopy is the first experiment that can be employed for the characterization of colloidal solutions containing nanoparticles of different types and shapes. The UV-Vis absorption spectrum of the colloidal solution containing silver nano stars is presented in Figure 1 together with the optical images of the colloids, right after synthesis (inset a) and after purification (inset b) through centrifugation, respectively.

The spectrum is characterized by the occurrence of a strong absorption peak located around 420 nm, specific for silver nanoparticles. The broadness of this peak is quite high with respect to isotropic silver nanoparticles [10,11,12], and this is a direct consequence of the presence of anisotropic silver nanostars in the solution. The full-width half-maximum (FWHM) value is ~200 nm in this case, which is more than double as compared to silver spherical nanoparticles. However, this property allows the use of NIR excitation lasers (785/830 nm) for SERS analysis of biological samples, strongly decreasing their intrinsic fluorescence and facilitating the recording of highly qualitative SERS spectra of pharmaceutical compounds, amino acids or even cell lysates.

### 2.2. Transmission Electron Microscopy (TEM) and Scanning Transmission Electron Microscopy (STEM) Characterization

TEM together with STEM has been employed for a proper characterization of AgNS. A typical TEM image of an individual nanostar is presented in Figure 2a, together with a zoom image of its central part (Figure 2b).

One can clearly distinguish that our AgNS consist of a central nanoparticle interconnected with several highly one-dimensional individual arms. The Energy Dispersive Analysis (EDS) confirmed that AgNS are composed of silver atoms (Appendix A). The STEM images of similar AgNS are presented in Figure 3.

The STEM images revealed the same structure as the one obtained from TEM images. The AgNS appear as branched 3D structures with high electron-scattering capacities. The structure points towards a {1 1 1} oriented crystalline structure [13] with a single center seed particle being the origin of the branching arms. Differences in the contrast of the outer section of the nanostars (Figure 3b) are also indicative of a {1 0 0} preferential growth of the crystals as presented in [14] and, thus, could be interpreted as having a 2D structure. These results are complementary to the TEM morphological analysis of the nanostars presented in Figure 2.

### 2.3. Atomic Force Microscopy (AFM) Characterization

The evolution of the AgNS deposited onto the heated ${\mathrm{MgF}}_{2}$ glass slides was evaluated by means of AFM measurements (Figure 4). All the measurements were performed on semi-contact operating mode. The 2D and 3D topographic images (height images) were recorded together with phase contrast ones (data not shown). The phase contrast images gave us a clear indication regarding the substrate’s composition which is a very homogenous one (very small phase variations have been detected on the samples). On the other hand, the topographic images confirmed the presence on the substrate of interconnected AgNS that coexist with individual ones (Figure 4a). Their sizes vary between a few hundred nanometers and a few microns. The cross-sectional analysis of the individual nanoflowers revealed they have a maximum height of 600 nm, which is much smaller (~one order of magnitude) than their lateral dimensions.

As a result of the slow heating process, the space between the individual one-directional arms (identified in the TEM images) has been completely filled, as can be noticed in the high-resolution topographic image presented in Figure 5. The formation of the 2D network has been also observed.

### 2.4. SERS and Raman Analysis

All SERS measurements included in this study were performed on plasmonic substrates obtained by pouring colloidal solutions containing silver nanostars on Raman transparent ${\mathrm{MgF}}_{2}$ glass slides heated at 50 °C. The as-obtained plasmonic substrates al-lowed the recording of very reproducible SERS spectra on different regions of the substrate for all the bioanalytes here presented.

All Raman measurements were performed on ${\mathrm{MgF}}_{2}\;$ transparent glass using a 785 nm laser excitation.

The SERS spectrum of the pristine silver substrate were recorded under the same experimental conditions and can be observed in Figure 6. The intensity of the vibrational bands is very low as can be seen in the figure, emphasizing, the capacity of the here-proposed plasmonic substrates to be employed for SERS analysis of various bioanalytes.

The enhancement capabilities of the plasmonic substrate were first tested on rhodamine 6G (R6G). This analyte was used in a first instance for testing the reproducibility of the recorded spectra. Finally, the Raman and SERS spectra of R6G were employed for the calculation of the substrate’s enhancement factor (EF). In the case of SERS, 1 μL solution of 1 mM R6G was deposited on the plasmonic substrate and left to dry. In the case of Raman, 1 μL solution of 1 mM R6G was poured on a ${\mathrm{MgF}}_{2}$ glass slide and left to dry. The SERS spectrum of R6G is presented in Appendix A. A value of ~8 × 10^3^ was calculated for the substrate’s enhancement factor (EF). This value was obtained by comparing Raman (Appendix A) and SER spectra of R6G (Appendix A).

For a proper assessment of the substrates’ plasmonic properties, we performed SERS measurements using as analytes a standard test molecule (methylene blue-MB) and three pharmaceutical compounds having different Raman cross sections: doxorubicin (DOX), atenolol (ATE), and metoprolol (MET). The capacity of our AgNS to perform as very efficient plasmonic substrates was further tested on cell lysates (DLD1 cells) and two amino acids (methionine and cysteine) The spectra were recorded using NIR excitation laser (785 nm).

In the case of MB and DOX, previous studies reported comprehensive analysis of their SERS spectra collected on different plasmonic substrates together with a complete vibrational band assignment. Given their great Raman cross section, they can be considered “standard” analytes for the evaluation of the plasmonic properties for new substrates [10,12,15]. The MB Raman and SERS spectra, collected on our substrates, are presented in Figure 7.

The major vibrational peaks of MB, as described in the literature, are clearly visible even for an analyte concentration of µM. The main vibrational bands of MB are located at 451, 504, 677, 773, 1178 and 1395 cm^−1^. The most visible ones can be assigned to C-N-C skeletal deformations (447 and 500 cm^−1^) and asymmetric C-N stretching (1395 cm^−1^). The assignment of the most important bands is presented in Appendix A. The spectrum is similar with the one recorded using other types of plasmonic substrates, suggesting a similar geometry of interaction between the substrate and the analyte.

The first pharmaceutical compound that was employed for testing the plasmonic substrate was doxorubicin. The typical vibrational frequencies of DOX are presented in Figure 8. The SERS spectrum of DOX included in this figure were collected using the individual nanostar highlighted in the inset of Figure 8. This spectrum is very similar to those reported in the scientific literature [16] recorded using other plasmonic substrates. The majority of DOX vibrational bands originate from the conjugated aromatic chromophore of the drug molecule. A tentative assignment of the most important bands is summarized in Appendix A.

In order to prove the capacity of the here-reported AgNS to be used as plasmonic substrates for biomolecules with low Raman cross section, we have performed SERS measurements on Metoprolol (Figure 9) and Atenolol (Figure 10). The spectra were recorded on an individual AgNS (MET) and on a solid substrate containing interconnected AgNS (ATE), respectively. The Raman spectra of MET and ATE are also presented (Figure 9b and Figure 10b, respectively).

In the scientific literature, there are very few studies reporting a SERS analysis of these two pharmaceutical compounds belonging to the class of beta-blockers [11]. These two drug molecules have a very similar chemical structure. The central part of both molecules consists of a phenyl ring linked to an amine alkanol side chain containing the asymmetric carbon atom.

As expected, both spectra included in Figure 9 and Figure 10 are very similar, being dominated by the bands assigned to the vibrational breathing mode of the central phenyl ring located at 852 cm^−1^ (MET) and 859 cm^−1^ (ATE), respectively. Other significant vibrational bands with their corresponding attributions are listed in Appendix A. According to Moskovits’ surface selection rules, this is a direct proof of a similar geometry of interaction between these two molecules and the plasmonic substrates [17]. On the other hand, the quality, reproducibility, and ease of recording of the SERS spectra can be attributed to the unique plasmonic properties of AgNS.

Very recently it has been shown that Raman/SERS spectroscopy has the potential to provide very useful information related to the nanoscale molecular interactions between analytes and plasmonic substrate [5,18].

As such, cell lysates were used for testing the plasmonic properties of our AgNS. A typical SERS spectrum of DLD1 cell lysate is presented in Figure 11a, while the Raman spectrum is presented in Figure 11b.

The spectrum from Figure 11 is dominated by a very intense vibrational peak at 725 cm^−1^. This peak can be assigned to the presence of adenine and adenosine as it has been recently shown by Genova et al. [19]. The presence of adenosine triphosphate in relatively high concentration could be an explanation for the strong intensity of this vibrational band. In the spectral fingerprint region (600–900 cm^−1^) one can detect the presence of another two distinct peaks at 627 cm^−1^ and 656 cm^−1^. The latest peak can be assigned to guanine ring breathing vibrational mode. This behavior represents a strong proof of the fact that, in the case of untreated cells, one can detect the presence of the same biomolecular species responsible for the occurrence of specific vibrational bands. According to Barhoumi et al. [20], the rest of the prominent vibrational bands observed in the lysate spectrum can be assigned to vibration modes of PO_2_ (1094 cm^−1^) and adenine (1328 & 1586 cm^−1^), respectively.

The last class of bio analytes that were used for testing the Raman signal enhancing capacities of our plasmonic AgNS were the amino acids. We have chosen Cysteine (Cys) and Methionine (Met) because their sidechain contains one sulfur atom that has the capacity to interact directly with the silver surface [21]. SERS and Raman spectra of Cys and Met are presented in Figure 12 and Figure 13, respectively.

The strong affinity of Sulphur atoms for silver surface is confirmed by the presence of two very intense vibrational bands (665 cm^−1^ for Cys & 682 cm^−1^ for Met) that have been assigned to C-S stretching vibrations of the two amino acid, in SERS spectra of Cys (Figure 12a) and Met (Figure 13a). Nevertheless, the presence of a strong vibrational peak around 1050 cm^−1^ (1054 cm^−1^ for Cys/1049 cm^−1^ in the case of Met) indicates a possible interaction of amino acids with the silver substrate through the amino group.

Appendix A represents a tentative assignment of the main vibrational bands for all the samples that were analyzed in this study.

## 3. Materials and Methods

### 3.1. Materials

All chemical compounds employed in this paper were of analytical grade. Silver nitrate (${\mathrm{AgNO}}_{3}$) and trisodium citrate were purchased from Roth (Karlsruhe, Germany) and Merck (Darmstadt, Germany), respectively. The aqueous solutions were prepared in Milli-Q water (Milli-Q® Direct Water Purification System, Darmstadt, Germany). ${\mathrm{MgF}}_{2}\;$ polished glasses (Crystran Ltd., Poole, UK), having a diameter of 20 mm, were used as port probes for the creation of the solid SERS substrates. The analytes and the pharmaceutical compounds used for SERS measurements (methylene blue MB, doxorubicin DOX, atenolol ATE, metoprolol MET, cysteine Cys and methionine Met) were purchased from Sigma-Aldrich (St. Louis, MO, USA). Cell culture products: Roswell Park Memorial Institute (RPMI 1640) cell culture medium, Fetal Bovine Serum (FBS), Glutamine, Penicillin/Streptomycin and Phosphate Buffered Saline solution (PBS 1X) were purchased from Gibco (Grand Island, NY, USA).

### 3.2. Cell Culture and Lysates

DLD1 (ATCC^®^ CCL-221™) colorectal carcinoma cells were acquired from American Type Culture Collection (ATCC, Manassas, VA, USA) and were maintained in RPMI 1640 cell culture medium, supplemented with 10% FBS, 1% Glutamine and 1% Penicillin/Streptomycin. During the experiment, the cells were stored in a humidified incubator at 37 °C and 5% CO_2_.

For obtaining the cell lysates, the cells were detached from the culture flasks and washed three times with fresh complete medium, followed by three washing steps with PBS 1X. Then the cell pellet was washed three times with UltraPure DNAse/RNAse free water. The washing steps were performed by centrifugation of the cells at 600× *g*, for 5 min at room temperature. After the washing procedures, the cell pellet was resuspended in 333 μL of UltraPure DNAse/RNAse free water and stored at −80 °C until further use.

The lysate processing involved mechanical lysis steps. The samples were removed from −80 °C and placed in the heating block at 37 °C for 15 min. The samples were sonicated with the following settings for the EpiShear 5/64′ (2 mm) Probe Sonicator (Active Motif, Carlsbad, CA, USA): 18 s with pulse of 3 s and 80% amplitude (from a maximum intensity level of 200 μm). After the sonication process, the samples were stored at −80 °C for 15 min, and the samples were sonicated and frozen two more times.

The cell lysate, processed as mentioned above, was stored at −80 °C and thawed only when the sample was analyzed.

### 3.3. Fabrication of Silver NanoStars (AgNS)

Colloidal suspensions of silver nanostars (AgNS) were prepared using an original method developed in MedFuture’s laboratories. The chemical reduction of ${\mathrm{Ag}}^{+}$ ions by trisodium citrate molecules were performed in a sealed-bottom flask introduced in the center of a microwave oven having a nominal power of 2100 W, operating at 2450 MHz capable of inducing water boiling (100 °C) after 30 s of irradiation. After 60 s of heating, the vials were removed from the oven, immersed in a water bath at 35 °C and let to accommodate at this temperature for 5 min, with no stirring. In a typical synthesis procedure, 10 mg of ${\mathrm{AgNO}}_{3\;}$ were dissolved in 50 mL of Milli-Q water and introduced in an empty 100 mL bottom flask. Trisodium citrate solution (1%, 1 mL) was added to the flask. The two solutions were gently stirred for 20 s, and then the bottom flask was sealed and introduced in the microwave oven. No sign of chemical reaction between the silver ions and the citrate molecules was observed at this stage. Afterwards, the solution was exposed to microwaves for 1 min. It was noted that, after 30 s of microwave exposure, the aqueous solution started to boil in the whole volume. After 15 more seconds, the solution suddenly changed its color from colorless to milky grey, indicating the successful formation of silver nanostars. By the end of the process, the colloidal solution was cooled down at room temperature using a two-step process. First the vials were immersed in a water bath at 35 °C for 5 min, and then they were brought to room temperature and left for another 5 min in air. The whole synthesis procedure took no more than 15 min.

### 3.4. Preparation of SERS Substrates

The SERS substrates were prepared by pouring 1.5 µL of as-synthesized AgNS solution onto a ${\mathrm{MgF}}_{2}\;$ glass slide heated at 50 °C. After 2 min from the deposition, the glass slides were removed from the heated plate and the substrates were let to accommodate at room temperature for 15 min. Once this final procedure was completed, the SERS substrates containing self-aggregated as well as individual nanostars were ready for use.

### 3.5. Preparation of Samples for SERS and Raman Investigation

Stock aqueous solutions of standard Raman-test molecules (methylene blue MB) as well as of therapeutic agents (doxorubicin DOX, atenolol ATE, metoprolol MET) and amino acids (cysteine Cys and methionine Met) were prepared. The concentration of all stock solution was 1 mM. Further dilutions were prepared using Milli-Q water. Once the drying process of the solid SERS substrates was completed, a very small volume of analytes aqueous solution (1 µL) was poured in the center of the solid substrate. The solution was left to dry for 30 min at room temperature. By the end of this process the samples were ready for SERS measurements, using a NIR excitation laser (785 nm).

The Raman spectra were recorded on dry solutions poured over Raman transparent ${\mathrm{MgF}}_{2}\;$ glass, using the same 785 nm laser excitation.

### 3.6. Characterization of Silver Nanostars and Solid Plasmonic Substrates

The colloidal solutions containing silver nanostars and the solid plasmonic sub-strates they formed were characterized by UV-VIS, STEM, TEM, AFM and SERS.

UV-Vis absorption measurements of the silver colloid were performed using a T92+ Spectrophotometer (PG Instruments, Lutterworth, UK) in the 190–900 nm wavelength range with a 2 nm resolution.

Electron microscopy measurements were performed on a Hitachi HT7700 Transmission Electron Microscope (Hitachi, Tokyo, Japan) operating at 120 kV, in the high-resolution mode.

The topography of the substrates was investigated by an Atomic Force Microscope (NT-MDT NTEGRA, NT-MDT Spectrum Instruments, Zelenograd, Russia) coupled with a micro-Raman system. The topographic images presented in this paper were acquired in the semi-contact operating mode.

The SERS spectra were recorded using a multilaser confocal Renishaw InVia Reflex Raman spectrometer. The wavelength calibration was performed on an internal reference sample (silicon). The 785 and 830 nm laser lines were employed as the excitation sources, but preliminary tests have been performed also on 532 and 633 nm laser lines. The emit-ting laser power was measured for each laser and for each objective on top of the sample. The SERS spectra presented in this paper were recorded using a 50× objective and different acquisition times ranging from few seconds to 50 s. The spectral resolution of the spectrometer was 0.5 cm^−1^. Baseline correction was applied to all SERS spectra, in order to eliminate the background. Each spectrum represents the average of a minimum of 30 spectral acquisitions. The baseline correction was performed by using the Wire 4.2 software provided by Renishaw (Gloucestershire, UK), with the inVia spectrometer. For all the spectra included in this study, the intensity of the vibrational bands is represented in kconts/(mW × s) units.

The calculations of EF have been performed according to the procedure developed by Gupta and Weimar [22], using the following equation:$$\mathrm{EF}=\frac{{\;M}_{\mathrm{Raman}}\times S_{\mathrm{Surf}}\times {\;I}_{\mathrm{Surf}}}{{\;M}_{\mathrm{Surf}}\times S_{\mathrm{Raman}}\times {\;I}_{\mathrm{Raman}\;}}$$
where M_Surf_ and M_Raman_ are the numbers of molecules, S_Surf_ and S_Raman_ are the geometrical areas of the molecular films and I_Surf_ and I_Raman_ are the SERS/Raman intensities of the most intense vibrational band that was used for the calculation of EF (1510 cm^−1^). Both measurements were performed using a 50× objective and an excitation laser of 785 nm. In the case of Raman measurements, a 100% laser power was used, the acquisition time was 10 s and the number of spectral acquisitions was 4. In the case of SERS measurement, the only modification that was made was the laser power which was set to 0.1%. The laser intensity, measured on sample surface, was 113 mW (100 % laser power) and 0.22 mW (0.1% laser power,), respectively. The intensities of all vibrational bands included in this study were plotted in kcounts/(mW × s) units. In the case of Raman/SERS measurements employed for EF calculation, 1 mM aqueous solutions of R6G were used. The diameter of the circular spot was ~2 mm in both cases. The Raman/SERS values of 1510 cm^−1^ band intensities were 43.3 and 0.0054 kcounts/(mW × s), respectively.

## 4. Conclusions

In this research, we report a very simple, rapid and reproducible microwave-aassisted method for the fabrication of anisotropic silver nanostars (AgNS) that can be successfully used as highly efficient SERS substrates for a broad range of bioanalytes: pharmaceutical compounds, cell lysates and/or amino acids. The AgNS consist of a central nanoparticle interconnected with several highly one-dimensional individual arms having a star shape. The microwave-assisted synthesis of the nanostars was performed in a sealed- bottom flask in a very short time interval (less than 3 min). The only chemical precursors employed in this wet-chemical procedure were silver nitrate and trisodium citrate, with no addition of any other reagents. Based on their unique properties, our plasmonic substrates have the capacity to generate very intense SERS spectra even in the case of using a NIR excitation laser, which represents a sine qua non condition for SERS measurements on biological samples.

The high quality and reproducibility of the vibrational spectra collected using the here-proposed plasmonic substrates represents strong experimental evidence of their versatility and application potential, especially in the case of bioanalytes.

## Figures and Tables

**Figure 1 ijms-23-08830-f001:**
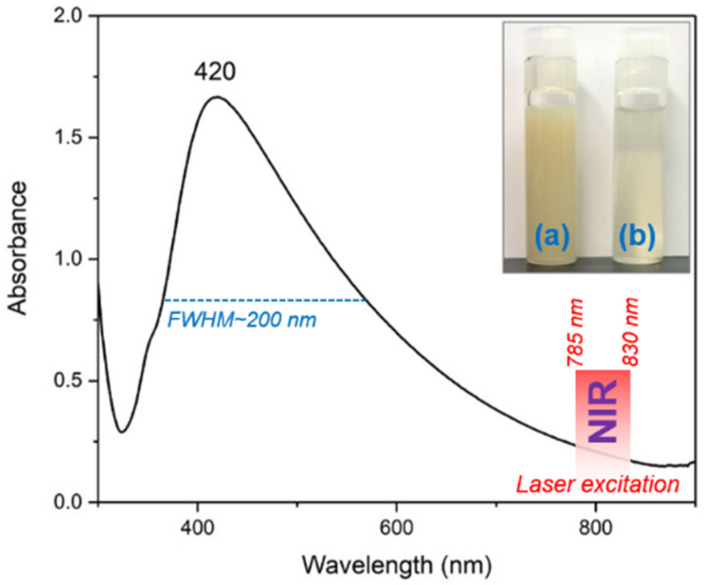
UV-Vis absorption spectra of the colloidal solution containing AgNS. The inset shows an optical image of the colloidal solutions before (**a**) and after purification through centrifugation (**b**) respectively.

**Figure 2 ijms-23-08830-f002:**
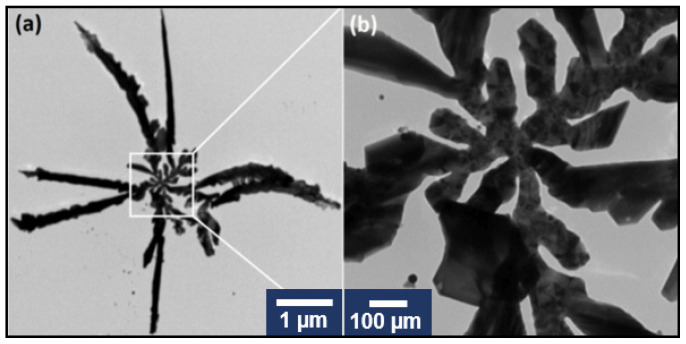
TEM image of a typical individual silver nanostar (**a**). TEM image of the central zone of the nanostar (**b**).

**Figure 3 ijms-23-08830-f003:**
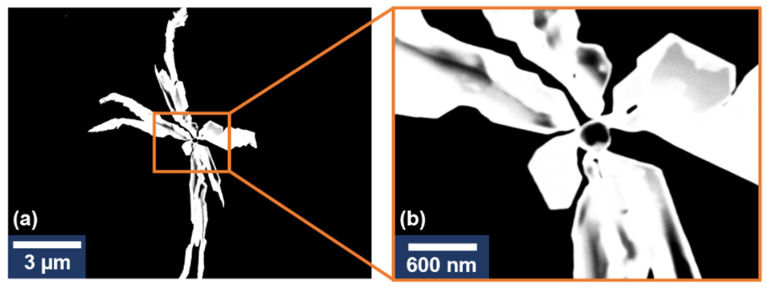
STEM image of an individual silver nanostar (**a**). STEM image of the central zone of the nanostar (**b**).

**Figure 4 ijms-23-08830-f004:**
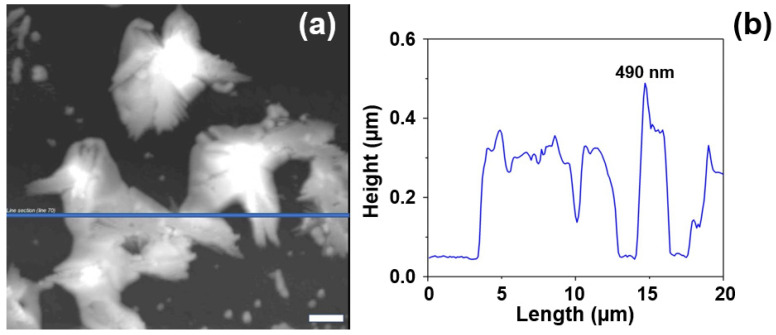
AFM topological image of several interconnected nanostars self-organized on the ${\mathrm{MgF}}_{2}\;$ surface (**a**). Corresponding cross-sectional analysis of AgNS (**b**). The scale bar represents 2 µm.

**Figure 5 ijms-23-08830-f005:**
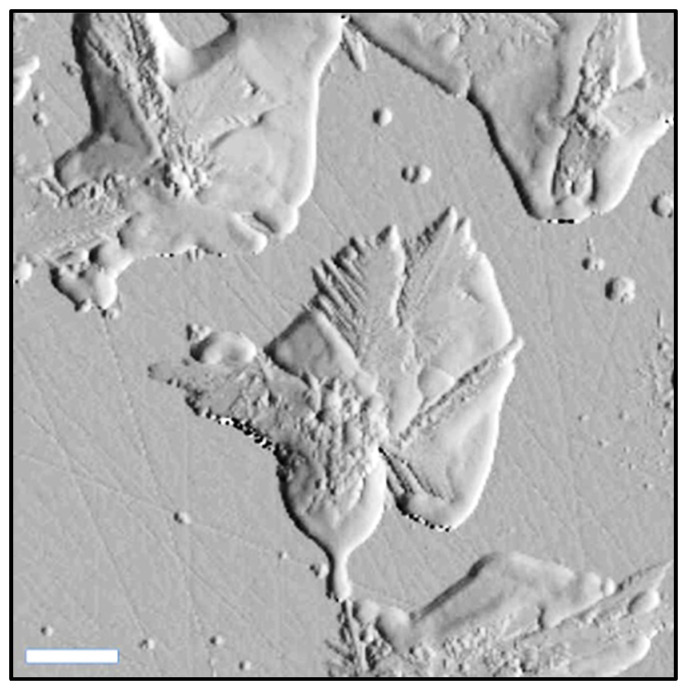
High resolution AFM topographical image of silver nanostars. The scale bar represents 2 µm.

**Figure 6 ijms-23-08830-f006:**
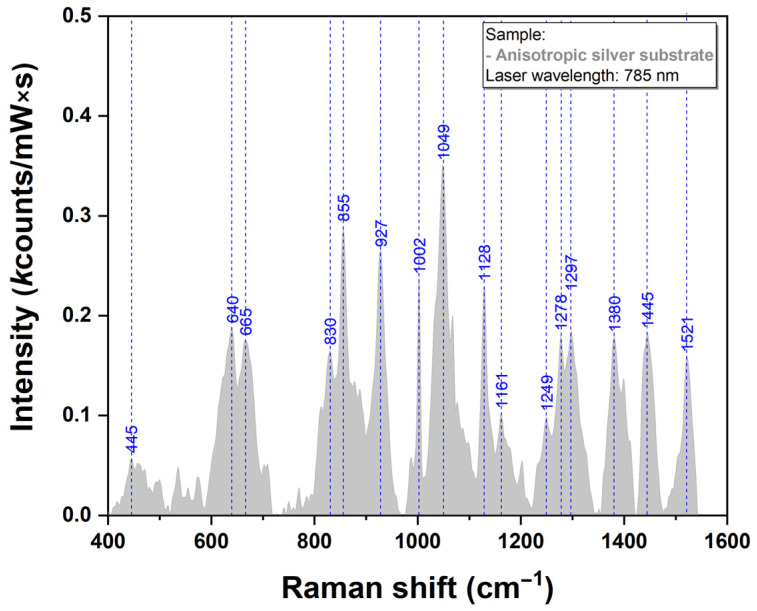
Raman spectra of AgNS recorded using a 785 nm laser excitation.

**Figure 7 ijms-23-08830-f007:**
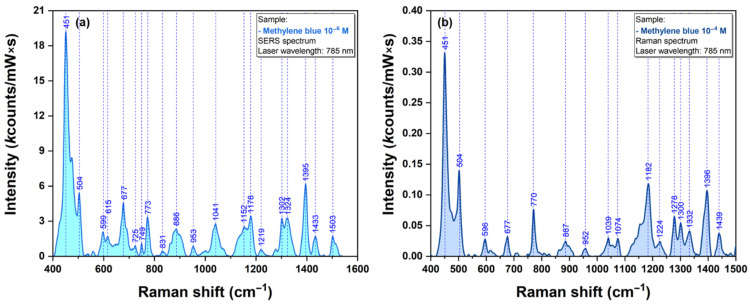
SERS spectrum of dried MB solution (1 µM) using a 785 nm excitation laser (**a**). Raman spectrum of dried MB solution (1 µM) using a 785 nm excitation laser (**b**).

**Figure 8 ijms-23-08830-f008:**
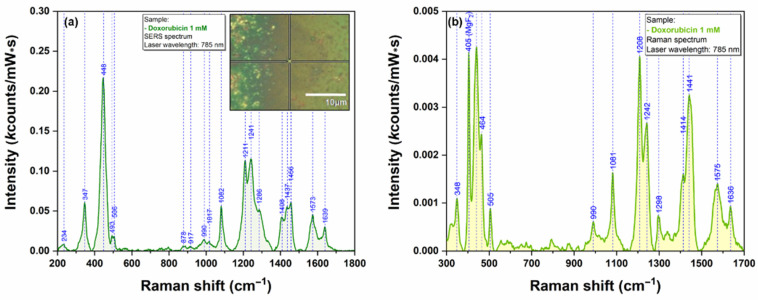
SERS spectrum of dried DOX solution (1 mM) collected on an isolated nanostar using a 785 nm excitation laser (**a**). The inset shows an optical image of the individual nanostar that has been used as plasmonic substrate. Raman spectrum of dried DOX solution (1 mM) collected using a 785 nm excitation laser (**b**).

**Figure 9 ijms-23-08830-f009:**
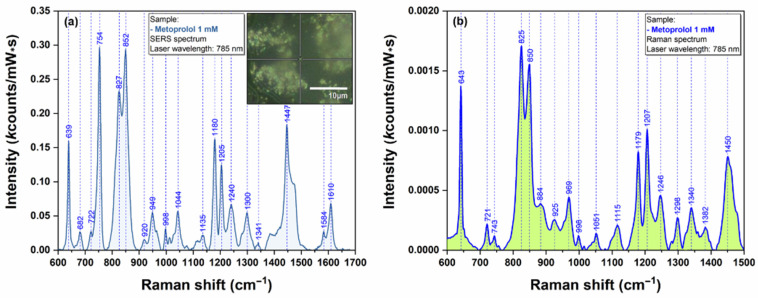
SERS spectrum of MET 1mM (**a**) collected on individual AgNS using a 785 nm laser. The inset in figure (**a**) shows an optical image of the individual nanostar that has been used for recording the spectra. Raman spectrum of MET 1 mM (**b**) collected using a 785 nm laser.

**Figure 10 ijms-23-08830-f010:**
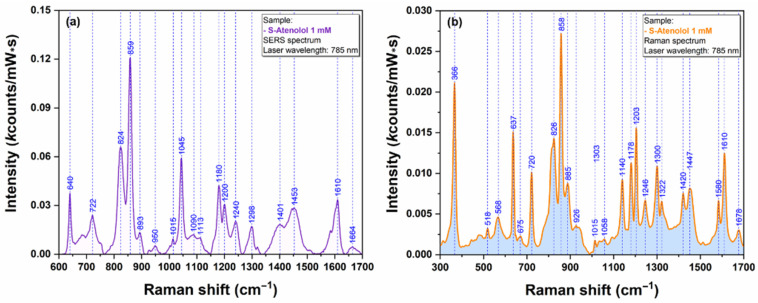
SERS spectrum of ATE 1 mM (**a**) collected on individual AgNS using a 785 nm laser. Raman spectrum of ATE 1 mM (**b**) collected using a 785 nm laser.

**Figure 11 ijms-23-08830-f011:**
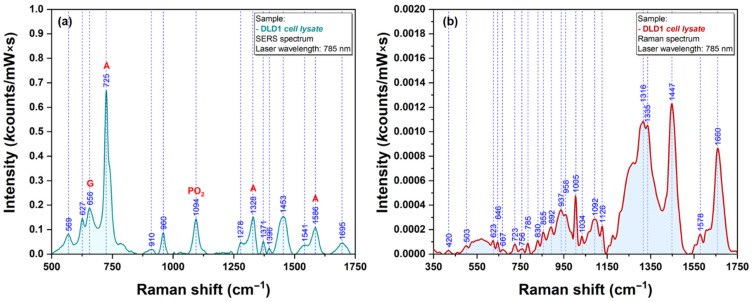
SERS spectra of DLD1 cell lysate obtained using a 785 nm excitation laser (**a**). Raman spectra of DLD1 cell lysate obtained using a 785 nm excitation laser (**b**).

**Figure 12 ijms-23-08830-f012:**
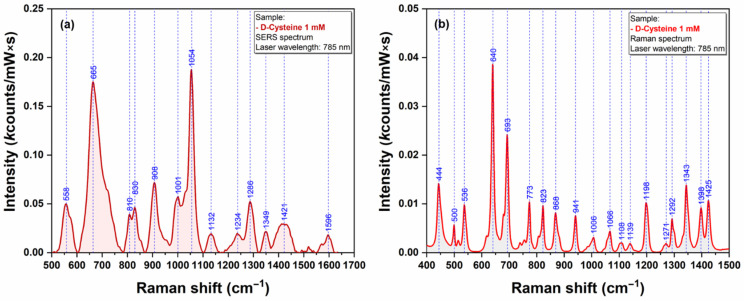
SERS (**a**) and Raman (**b**) spectra of Cysteine (1 mM) recorded using a 785 nm excitation laser.

**Figure 13 ijms-23-08830-f013:**
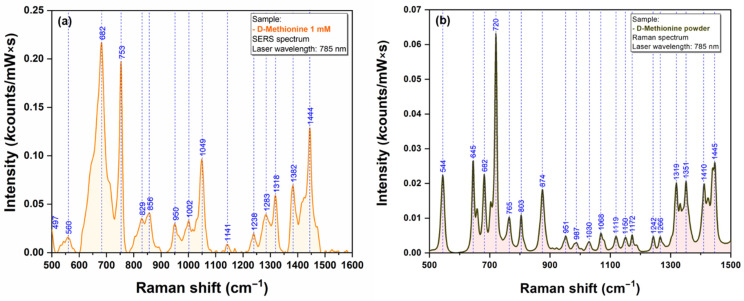
SERS (**a**) and Raman (**b**) spectra of Methionine (1 mM) (**a**) recorded using a 785 nm excitation laser.

## Data Availability

Not applicable.

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
