# Peer review of "Facile Microwave Assisted Synthesis of Silver Nanostars for Ultrasensitive Detection of Biological Analytes by SERS"

_ijms, 2022, doi:10.3390/ijms23158830_

Round 1
Reviewer 1 Report
The Review Report is attached.

Author Response
Dear reviewer,
Thank you for your time and suggestions in improving the scientific quality of our manuscript. In the next paragraph you can find our point by point answers to your comments:
- The microwave operating system was capable to induce water boiling (100°C) after 30 s of irradiation. We included this detail in the manuscript at Materials and Methods section.
- We have replaced figures 2, 4b, 6, 7, 8a, 8b and 9 with others of an increased quality. Also, we have included Raman spectra for all the samples analyzed in our study.
- We have modified the scale of Figures 2a and 2b at your recommendations.
We highlighted in yellow all the modifications we have made.
Yours sincerely,
Prof. Rares Ionut STIUFIUC
Reviewer 2 Report
The article by Revnic and coworkers explores in detail a new silver nanostar synthesis method via chemical reduction of silver ions with trisodium citrate. Silver nanostars used in SERS measurements of methylene blue, doxorubicin, atenolol, metoprolol, methionine and cysteine. Although it has a low SERS enhancement factor compared to well-developed plasmonic particles, it is a promising one-step synthesis method.
Comments:
1. The authors mentioned about the Energy Dispersive Analysis (EDS) in the manuscript, but they did not show the data. The data should be given in Supporting Information.
2. Raman spectra of pristine silver nanostars should be given in Supporting Information.
3. The detail information on page 6, lines 184-192 can be given in Material and Method part.
4. The Raman spectra of MB, DOX, MET, ATE, DLD1 cell lysate, cysteine, methionine should be included in the manuscript for an efficient comparison of their Raman and SERS spectra.
5. A Raman Band Assignment table should be included in the manuscript.
Author Response
Dear Reviewer,
Thank you for your time and recommendations for improving the scientific quality of our manuscript. We highlighted in yellow all the modifications that we have made to the manuscript, according to your suggestions. Here are the point-by-point answers to your questions:
- Thank you for your suggestion regarding EDS analysis. Accordingly we included EDS data in the supplementary files (figure S1).
- We included the Raman spectrum of pristine silver nanostars in the manuscript at results section.
- Very good observation. We have moved the information from page 6 lines 175-191 to the Materials and Methods section.
- We inserted a Raman Band Assignment table in the manuscript supplementary files (Table S1).
The SI file has been also uploaded in the system.
Thank you again for your time and effort
Yours sincerely,
Prof. Rares Ionut STIUFIUC
Round 2
Reviewer 2 Report
The authors have made all the revisions according to the comments. The manuscript can be published in this present form.
Sincerely,
Dr. Seda Kelestemur